# Atrial Fibrillation and the Risk of Ventricular Arrhythmias and Cardiac Arrest: A Nationwide Population-Based Study

**DOI:** 10.3390/jcm12031075

**Published:** 2023-01-30

**Authors:** Ameenathul M. Fawzy, Arnaud Bisson, Alexandre Bodin, Julien Herbert, Gregory Y. H. Lip, Laurent Fauchier

**Affiliations:** 1Liverpool Centre for Cardiovascular Science at University of Liverpool, Liverpool John Moores University and Liverpool Heart & Chest Hospital, Liverpool L14 3PE, UK; 2Service de Cardiologie, Centre Hospitalier Régional Universitaire et Faculté de Médecine de Tours, 2 Boulevard Tonnellé, 37000 Tours, France; 3Cardiology Department, Centre Hospitalier Régional d’Orléans, 45067 Orléans, France; 4Department of Clinical Medicine, Aalborg University, 9000 Aalborg, Denmark

**Keywords:** atrial fibrillation, sudden death, cardiac arrest, ventricular tachycardia, ventricular fibrillation, arrhythmias

## Abstract

Background: Atrial fibrillation (AF) has been linked to an increased risk of ventricular arrhythmias (VAs) and sudden death. We investigated this association in hospitalised patients in France. Methods: All hospitalised patients from 2013 were identified from the French National database and included if they had at least 5 years of follow-up data. Results: Overall, 3,381,472 patients were identified. After excluding 35,834 with a history of VAs and cardiac arrest, 3,345,638 patients were categorised into two groups: no AF (*n* = 3,033,412; mean age 57.2 ± 21.4; 54.3% female) and AF (*n* = 312,226; 78.1 ± 10.6; 44.0% female). Over a median follow-up period of 5.4 years (interquartile range (IQR) 5.0–5.8 years), the incidence (2.23%/year vs. 0.56%/year) and risk (hazard ratio (HR) 3.657 (95% confidence interval (CI) 3.604–3.711)) of VAs and cardiac arrest were significantly higher in AF patients compared to non-AF patients. This was still significant after adjusting for confounders, with a HR of 1.167 (95% CI 1.111–1.226) and in the 1:1 propensity score-matched analysis (*n* = 289,332 per group), with a HR of 1.339 (95% CI 1.313–1.366). In the mediation analysis, the odds of cardiac arrest were significantly mediated by AF-associated VAs, with an OR of 1.041 (95% CI 1.040–1.042). Conclusion: In hospitalised French patients, AF was associated with an increased risk of VAs and sudden death.

## 1. Introduction

Atrial fibrillation (AF) is a multifaceted condition with layers of complexity. The onset of this arrhythmia is associated with several complications and regarded as a marker of poor prognosis and predictor of mortality [1]. Mortality associated with AF is 1.5–2-fold higher with one study reporting risk as high as 4-fold compared to patients without AF [2].

The leading cause of death in AF patients is predominantly heart failure, with ischaemic heart disease and sudden death amongst the others [3,4]. Although sudden death in AF was believed to be related to co-morbidities such as heart failure, emerging evidence suggests that AF may be independently associated with an increased risk of ventricular arrhythmias (VA) and cardiac arrest [5]. This relationship between AF and sudden death was demonstrated about a decade ago in the Atherosclerosis Risk in Communities (ARIC) study and the Cardiovascular Heart Study (CHS), which indicated a 2.5 times higher risk of sudden death in those with incident AF [6].

In post hoc analyses of the hallmark RE-LY trial which compared dabigatran versus warfarin and the ENGAGE AF-TIMI 48 trial that compared edoxaban versus warfarin in AF patients, sudden death accounted for 22% and 32% of all deaths, respectively [7,8]. Sudden death characterisation in the latter study revealed that documented arrhythmias accounted for just 11% of sudden death events, with VAs in an even smaller proportion (4.9%). However, data were obtained retrospectively: only a limited number had post-mortem evaluations, and these studies are likely to have been underpowered [7].

The AF-VA relationship is poorly understood due to a paucity of cause-specific mortality data, lack of studies focusing on this subject as well as shared risk factors such as heart failure that also increase the risk of arrhythmias and sudden death. To date, the largest studies specifically evaluating the association between AF, VAs and sudden death are based on Asian populations. Therefore, in this study, we aimed to examine this association in a large European cohort.

## 2. Materials and Methods

### 2.1. Study Design and Population

This retrospective, observational cohort study was conducted across the French population of 67 million. Study participants were identified from the National Hospital database, *Programme de Médicalisation des Systèmes d’Information* (PMSI), which is an administrative database that includes medical information on every individual admitted to hospitals across the nation, accounting for over 98% of the French population. The system holds data pertaining to all hospitalisations including patient demographics, clinical presentations, diagnoses as well as procedures undertaken. Patient details are anonymised and recorded at the time of admission, and diagnoses are coded at discharge as per the International Classification of Diseases, Tenth Revision (ICD-10). Procedure details were obtained from the Classification Commune des Actes Médicaux (CCAM), which is a separate system that holds details of procedures according to the national nomenclature.

All patients seen in French hospitals in the year 2013 with at least 5 years of follow-up data were included in the study. Exclusion criteria included a history of ventricular tachycardia (VT), ventricular fibrillation (VF) and cardiac arrest (Figure 1). Diagnoses and outcomes were identified using respective ICD-10 codes.

As the study was retrospective and conducted using anonymised data without direct patient involvement, ethical approval was not required. Access to PMSI data was granted by the French Data Protection Authority, and procedures for handling the data were approved by the independent national ethics committee. All study proceedings were carried out according to the Declaration of Helsinki.

### 2.2. Follow-Up and Outcomes

Baseline was the date of first admission to hospital in 2013 and follow-up was continued until 31 December 2019 or death/emigration. The median follow-up duration was 5.4 years (interquartile range (IQR) 5.0–5.8 years).

Primary outcome was the combined endpoint of VT, VF or cardiac arrest (VT/VF/CA) during the follow-up period. This was recorded on the PMSI website with the corresponding ICD-10 codes (I472 for VT, I490 for VF and I46 for cardiac arrest), as adjudicated by the clinicians during hospitalisation. Patients with cardiac arrest included those who were successfully resuscitated and those who died.

### 2.3. Statistical Analysis

Continuous variables were expressed as mean (± standard deviation), and categorical variables were expressed as counts and percentages. Patients were categorised into 2 groups (AF and no AF) depending on the presence or absence of AF. Differences between groups were assessed with the *t*-test for continuous variables and chi-squared χ^2^ test for categorical variables.

Incidence rates of VT/VF/CA were expressed in person-time (years) and percentage (%)/year. Cox regression models were used to evaluate the association between covariates and study outcomes and to identify independent predictors of the outcomes. Results were expressed as hazard ratios (HR) with 95% confidence intervals (CI). Hazard ratios were also used to examine the association between AF and VT/VF/CA. Model A was unadjusted, model B was adjusted for age and sex, and model C was adjusted for the covariates included in Table 1.

To account for the large differences in baseline characteristics between the two groups and potential confounding effects, propensity score matching (PSM) was performed with patients in each group matched on a 1:1 basis, using the one-to-one nearest neighbour method (with a calliper of 0.01 of the standard deviation (SD) of the propensity score on the logit scale). Propensity scores were calculated using covariates in Table 1 with AF as the dependent variable. Standardised differences, calculated as the difference in the means or proportions of a particular variable divided by the pooled estimate of SD for that variable, were used to examine the distribution of demographic data and comorbidities in the two groups. A standardised difference of <5% was regarded as a marginal difference.

A mediation analysis was also carried out to further evaluate the total, indirect and direct effects of AF on cardiac arrests mediated by VT/VF, using the approach described by Baron et al. and Pearl et al. [9,10]. Results were expressed as odds ratios (OR) and 95% CIs.

All tests were 2-tailed and *p*-values of ≤0.05 were considered statistically significant. Statistical analysis was performed using Enterprise Guide 7.1, (SAS Institute Inc., SAS Campus Drive, Cary, NC, USA) and STATA version 16.0 (Stata Corp, College Station, TX, USA).

## 3. Results

A total of 3,381,472 patients who had at least 5 years of follow-up data were identified. Of these, 35,834 were excluded due to a prior history of VAs and cardiac arrest. The remaining 3,345,638 patients were categorised into two groups; no AF (*n* = 3,033,412; mean age 57.2 ± 21.4; 54.3% female) and AF (*n* = 312,226; 78.1 ± 10.6; 44.0% female) accounting for 90.8% and 9.2% of the study population, respectively (Figure 1).

The AF cohort represented a higher risk population with a significantly greater proportion of patients with risk factors such as hypertension, dyslipidaemia, diabetes, coronary artery disease (CAD), heart failure and valve disease. Of the baseline characteristics evaluated, only smoking and alcohol-related disorders had comparable sample prevalence. Patients with AF were also more likely to be male, significantly older and frailer, with a mean frailty index that was at least twice as high (Table 1).

### 3.1. Incidence of VT/VF/CA

During a median follow-up duration of 5.4 years (IQR 5.0–5.8 years), 105,151 events were observed across both groups. The incidence of VT/VF/CA was 0.56%/year (95% CI 0.56–0.57) in the no AF group and 2.23%/year (95% CI 2.20–2.26) in the AF group (Table 2), which is a 4 times higher rate in the latter group (Figure 2).

### 3.2. Predictors of VT/VF/CA in the Hospitalised Population

In the univariate analysis, all variables included were associated with an increased risk of VAs and cardiac arrest. After multivariable analysis, age, male sex, hypertension, diabetes mellitus, heart failure, history of pulmonary oedema, valve disease, dilated cardiomyopathy, CAD, vascular disease, AF, smoking, alcohol-related disorders, chronic kidney disease, lung disease, liver disease, inflammatory diseases, anaemia, previous cancer, poor nutrition and frailty were independently associated with an increased risk of the study outcomes (Table 3).

Male sex and heart failure were the strongest predictors of VT/VF/CA with more than 1.5 times the risk. Prior history of MI and percutaneous coronary intervention (PCI) were no longer statistically significant. In addition, dyslipidaemia was negatively associated with the incident outcomes, HR 0.876 (0.862–0.891), *p* < 0.01, suggesting its presence may paradoxically be associated with a lower risk of VAs and cardiac arrests.

### 3.3. Risk of VT/VF/CA

The risk of VT/VF/CA was significantly higher in patients with AF compared to those without AF, HR 3.657 (95% CI 3.604–3.711). After adjusting for age and sex (model B), this increased risk was attenuated but still significant: adjusted HR (aHR) was 2.166 (95% CI 2.133–2.199). Model C revealed an aHR of 1.349 (95% CI 1.327–1.372), demonstrating a 35% higher risk of VT/VF/CA in individuals with AF compared to those without (Table 4).

### 3.4. PSM Analysis

A total of 578,664 patients were included in the PSM analysis, with 289,332 patients in each group. Following this, the standardised difference between the two groups for each variable was <5%, and hence, marginal (Appendix A). In this matched cohort, the HR for incident outcomes was 1.339 (95% CI 1.313–1.366), indicating that the risk was still significantly elevated, with a 34% higher risk of VT/VF/CAs in the AF group.

### 3.5. Mediation Analysis

The total effect of AF on cardiac arrest was OR 1.594 (95% CI 1.564–1.624), whilst the natural direct effect (due to AF amongst patients with no VT/VF during FU) was OR 1.532 (95% CI 1.503–1.562) and the natural indirect effect (due to AF for patients that had VT/VF during FU) was OR 1.041 (95% CI 1.040–1.042). These results suggest that the association between AF and cardiac arrests is significantly mediated by the effect of AF on VAs.

## 4. Discussion

In this study, our findings indicate that (i) VT/VF/CA are 4-fold more common in hospitalised AF patients compared to non-AF patients; (ii) male sex and heart failure are the strongest predictors of VT/VF/CA, while the presence of dyslipidaemia may be associated with a lower risk; (iii) AF patients have a 35% higher risk of VT/VF/CA compared to the general population without AF—observed to a similar degree in the PSM analysis, and (iv) the risk of cardiac arrests is significantly mediated by VAs due to AF. These findings support existing evidence which allude to the increased risk of VAs and sudden death associated with AF.

To date, few studies have evaluated the risk of VAs and sudden death in the general population with AF. AF was first described as an independent predictor of sudden death by Chen et al. in post hoc analyses of the ARIC and CHS studies, which demonstrated a meta-analysed HR of 2.47 (95% CI 1.95–3.13) for sudden death [6]. Unfortunately, data on cause-specific mortality were unavailable. Bardai et al. conducted a population-based case-control study that suggested that AF was independently associated with a 3-fold risk of VF, regardless of the use of anti-arrhythmic drugs, presence or absence of MI or comorbidities, suggesting there are other contributory mechanisms at play [11].

In the largest study evaluating the association between Vas and AF, individuals with new onset AF in the Korean general population were found to have a 4.6-fold increased risk of Vas [12]. This figure is nearly 3.5 times higher than our reported aHR of 1.34, bearing in mind our primary outcome was a composite of Vas as well as cardiac arrests as opposed to Vas only. This difference may be due to important differences in the study design and population characteristics. Even though the study by Kim et al. had a larger population size, most patients belonged to the non-AF group, with AF patients comprising 0.13% of the total study population. Patients with prevalent AF, heart failure, ischaemic stroke, Vas and premature ventricular contractions were excluded from this study, in contrast to our study which only excluded those with a history of Vas and cardiac arrests. While the former approach will have minimised confounding effects to a larger degree, the yield of outcome events was significantly smaller in both the AF and non-AF groups [12]. Large differences in the sample size and outcome events with significant differences in baseline characteristics between the two groups were also a feature of our study; hence, a PSM analysis was carried out to address this.

Existing studies, including ours, have demonstrated an attenuation in the strength of association between AF and incident outcomes with the increase in the number of confounders adjusted for [13]. In the study by Kim et al., adjustments were made for a few confounders such as age, sex, BMI, smoking status, alcohol consumption status, physical activity status, hypertension, diabetes mellitus, and dyslipidaemia, as opposed to our study where rigorous adjustments were made for several variables. In a similar study based on the Taiwanese population, the aHR for a composite of VAs and cardiac arrests was comparable at 1.64, over a follow-up duration of 4.15  ±  3.78 years [14].

In our analysis, male sex and heart failure were the strongest predictors of VT/VF/CA While heart failure has strongly been associated with VAs, the evidence for a gender-based association in the context of AF is limited [4,15,16]. Despite the lifetime risk of AF being equivalent in both men and women, it is well known that female sex is associated with poorer AF-associated outcomes such as stroke/ thromboembolism and possibly even mortality. This is thought to be secondary to a myriad of factors ranging from the uptake of oral anticoagulation and other treatment modalities to sex differences in atrial electrophysiology and sex hormones [17]. Although this is the case, several observational and registry studies have demonstrated a significantly lower risk of VAs and sudden cardiac arrest in women compared to men, even after accounting for predisposing factors such as CAD and heart failure [18,19]. This may be because women are more likely to have structurally normal hearts and therefore less of an arrhythmogenic substrate for triggering VAs. A lower susceptibility to VAs due to genetics, proteomics and sex hormones such as oestrogen, influencing the electrophysiological properties and autonomic function of the heart, have also been proposed as potential mechanisms [19,20]. Nevertheless, the underlying processes for these sex differences are poorly understood, and further studies are warranted.

Interestingly, previous MI and PCI were not significantly associated with the incident outcomes, with the latter showing a trend towards a reduced risk, although not significant, with a HR of 0.982 (95% CI 0.953–1.011). This may be explained by the fact that the timely treatment of MIs with PCI may prevent ventricular scarring, thereby averting the formation of an arrhythmogenic substrate for VAs. Furthermore, these patients are likely to be commenced on treatments such as beta-blockers and under regular cardiology follow-up, which may reduce the risk of such arrhythmic events and secondary complications.

Notably, in our analysis, dyslipidaemia was significantly associated with a reduced risk of VT/VF/CA. While this seems anomalous, a similar finding was observed by Kim et al. where a significant interaction effect was noted between AF and dyslipidaemia as well as AF and age, suggesting that younger people without dyslipidaemia were at a greater risk of VAs due to AF [12]. Whether this is a true association or if there is a further interaction effect with statin use is unclear due to the lack of data on drugs in these database-based analyses. Nonetheless it does raise the question as some studies suggest that statins may exhibit a protective effect against AF and VAs owing to their pleiotropic effects such as membrane stabilisation, anti-inflammatory properties and prevention of adverse myocardial remodelling [21,22].

Despite the evolving hypothesis that AF increases the risk of VAs and sudden death, what is unclear is if AF is truly proarrhythmic at an intrinsic, cellular level or if the observed effects are mediated by the presence of other factors such as heart failure. In our study, even after adjustments were made for conditions such as heart failure, the VT/VF/CA risk was still elevated in the AF group, echoing findings from previous studies [11,12,14]. The association between AF and cardiac arrest was examined separately in the mediation analysis, indicating that AF-associated VAs significantly mediated the risk of cardiac arrests. However, the odds ratio for the direct effect suggests that there are other underlying processes, yet to be identified perhaps, influencing this relationship. Indeed, unmeasured confounders may also influence these observations, and this includes undetected CAD which may not necessarily manifest typically and therefore may not be known about. Studies evaluating angiographic profiles of AF patients without a known history of CAD have demonstrated an atherosclerotic CAD prevalence of up to 70%, suggesting the presence of AF as a marker of advanced coronary atherosclerosis [23,24]. This close association between AF and undetected CAD may be an important contributor to the risk of VT/VF/CA.

One postulated mechanism for a direct link between AF and VAs is a genetic component giving rise to a joint pathway for both these arrhythmias. Single nucleotide polymorphisms on chromosome 4q25 that are strongly associated with AF have been found to increase the risk of sudden death, while variants of the SCN5A and SCN10A genes have independently been associated with both AF and VF, although a direct association is yet to be determined [25,26]. Rapidly conducted AF can also give rise to broad QRS complexes which can lead to an R-on-T phenomenon: a common culprit of ventricular arrhythmogenesis, leading to VT/VF, and subsequently asystole [27]. The abrupt changes in the cycle length and reduction in the ventricular refractory period owing to irregular, rapid ventricular rates as well as autonomic activation of the heart associated with AF may increase arrhythmogenicity and play a role in inducing VAs [28,29,30]. Anti-arrhythmic drugs such as propafenone commonly used in AF have also been shown to increase the risk of VAs, although the incidence of this is regarded as low [15,16].

Further studies are required to determine whether a causal association exists between AF and VAs, with emphasis on cause-specific mortality data. This may require prospective studies and data linkage across different entities such as ambulance services and out-of-hospital cardiac arrest outcomes registries. This is imperative, as the presence of a direct link between AF and VAs may mandate the re-evaluation of our clinical practice with priority shifting towards establishing sinus rhythm to prevent lethal VAs and reduce risk of sudden death. Existing studies demonstrate several benefits of rhythm control strategies despite high rates of AF recurrence [31,32]. However, studies specifically evaluating whether the restoration of sinus rhythm lowers the risk of VT/VF/CA are currently lacking and warranted as evidence for these AF-associated complications emerges.

### Limitations

As our analysis was based on administrative data collected retrospectively, it is subject to the limitations of any such study. Data were retrieved using ICD-10 codes, so the accuracy was reliant on correct and comprehensive coding. Hence, a degree of information bias may exist due to potential coding inaccuracies. For example, although non-sustained VT episodes are more commonly coded as I49.3, some of these may have been encoded with I47.2 for VT. However, this is likely a small proportion due to measures taken to maintain data quality, and effects are minimal, given the large scale of the study population. Similarly, although ICD codes for AF subtypes (paroxysmal, persistent and permanent) have been specified as a subsection of I48 (I48.0, I48.1 and I48.2, respectively) over the recent years, they are not commonly utilised and only used in about 10% of the cases. For this reason, and due to AF subtypes not being as well distinguished by physicians who are not experts in arrhythmia, the codes for AF patterns were deemed possibly unreliable and therefore not used. Hence, it was not possible to determine the number of patients or analyse differences in risk between paroxysmal, persistent and permanent AF subgroups.

Furthermore, even though adjustments were made for several factors, confounding effects from those variables that were not considered will still exist. This includes medications such as anti-arrhythmic agents, rate control agents and statins, which have all been implicated in the incidence or recurrence of arrhythmias. Given the observational nature of the study, it was not possible to measure the AF burden and evaluate how it may have influenced outcomes. Additionally, VT/VF/CA numbers will have been underestimated, as these data were obtained from the PMSI database, which is based on hospital admissions, excluding those who experienced these events in the community, who will never have been admitted. Lastly, though our data suggest that AF is associated with an increased risk of VT/VF/CA, a causal inference cannot be drawn.

## 5. Conclusions

In this large, nationwide study of hospitalised patients, we demonstrate that AF is associated with an increased risk of VAs and cardiac arrests. The presence of a direct link between AF and VAs may elevate the importance of establishing sinus rhythm, but further studies are required to confirm these findings.

## Figures and Tables

**Figure 1 jcm-12-01075-f001:**
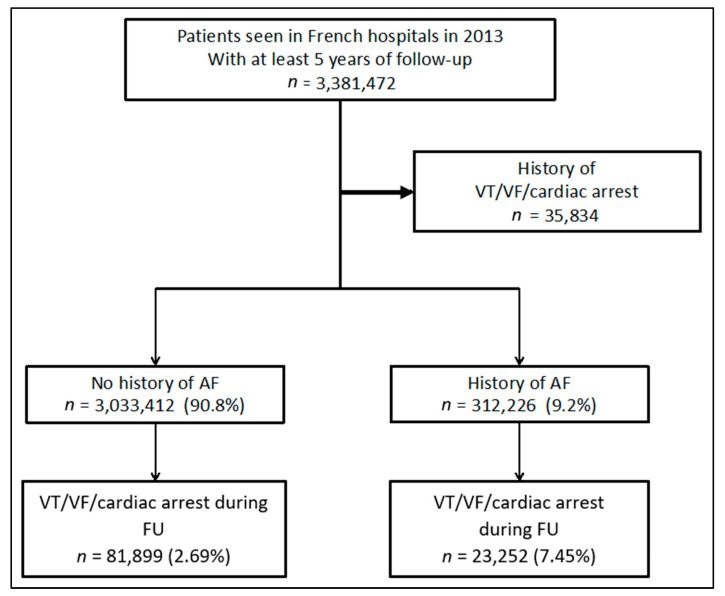
Study selection process. Abbreviations: AF—atrial fibrillation; FU—follow-up, VF—ventricular fibrillation; VT—ventricular tachycardia.

**Figure 2 jcm-12-01075-f002:**
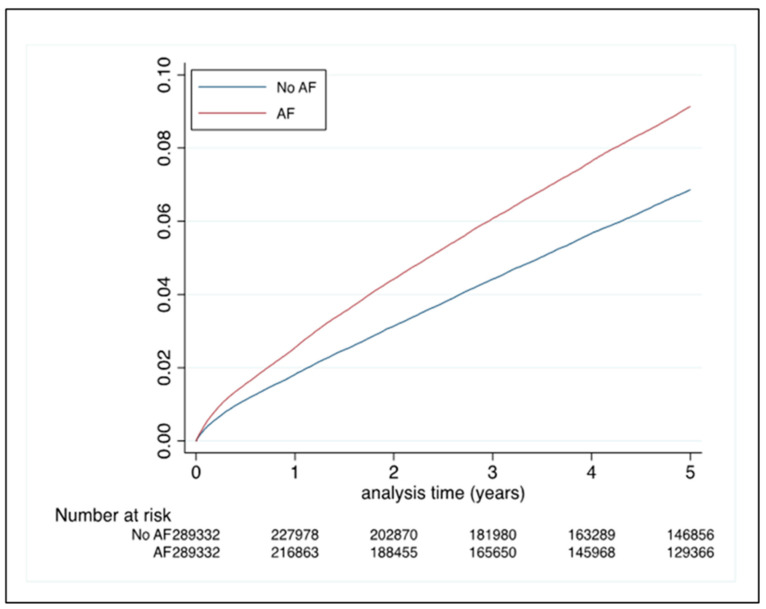
Incident VT/VF/CA in the matched cohorts. Abbreviations: CA—cardiac arrest, VF—ventricular fibrillation, VT—ventricular tachycardia.

**Table 1 jcm-12-01075-t001:** Baseline characteristics of the French hospitalised patients with and without AF.

Baseline Characteristics	Total	Before Matching			After Matching		
		No AF	AF	SDi *	No AF	AF	SDi
	(*n* = 3,345,638)	(*n* = 3,033,412)	(*n* = 312,226)	(%)	(*n* = 289,332)	(*n* = 289,332)	(%)
Age (years), mean ± SD	59.1 ± 21.5	57.2 ± 21.4	78.1 ± 10.6	124.0	77.7 ± 10.8	77.6 ± 10.6	−0.7
Sex (male), *n* (%)	1,562,002 (46.7)	1,387,154 (45.7)	174,848 (56.0)	20.7	164,634 (56.9)	161,616 (55.9)	−2.1
Hypertension, *n* (%)	1,014,316 (30.3)	799,376 (26.4)	214,940 (68.8)	94.0	203,184 (70.2)	193,786 (67.0)	−7.2
Diabetes mellitus, *n* (%)	460,626 (13.8)	378,657 (12.5)	81,969 (26.3)	35.4	78,225 (27.0)	74,766 (25.8)	−3.1
Heart failure, *n* (%)	335,808 (10.0)	188,903 (6.2)	146,905 (47.1)	104.1	120,800 (41.8)	124,229 (42.9)	3.0
History of pulmonary oedema, *n* (%)	20,929 (0.6)	11,523 (0.4)	9406 (3.0)	20.5	6942 (2.4)	7460 (2.6)	1.4
Valve disease, *n* (%)	116,586 (3.5)	63,329 (2.1)	53,257 (17.1)	52.6	37,679 (13.0)	39,514 (13.7)	2.2
Dilated cardiomyopathy, *n* (%)	72,025 (2.2)	38,617 (1.3)	33,408 (10.7)	40.5	24,757 (8.6)	26,204 (9.1)	2.2
Coronary artery disease, *n* (%)	345,007 (10.3)	251,195 (8.3)	93,812 (30.0)	57.5	87,299 (30.2)	82,429 (28.5)	−4.5
Previous MI, *n* (%)	51,624 (1.5)	38,985 (1.3)	12,639 (4.0)	17.2	12,328 (4.3)	11,354 (3.9)	−2.1
Previous PCI, *n* (%)	83,454 (2.5)	67,151 (2.2)	16,303 (5.2)	15.9	16,720 (5.8)	15,153 (5.2)	−2.9
Vascular disease, *n* (%)	279,140 (8.3)	211,598 (7.0)	67,542 (21.6)	42.8	62,815 (21.7)	59,747 (20.6)	−3.1
Ischemic stroke, *n* (%)	62,335 (1.9)	41,508 (1.4)	20,827 (6.7)	27.2	17,669 (6.1)	17,244 (6.0)	−0.8
Intracranial bleeding, *n* (%)	34,054 (1.0)	26,095 (0.9)	7959 (2.5)	13.1	7074 (2.4)	6888 (2.4)	−0.5
Smoker, *n* (%)	227,270 (6.8)	205,249 (6.8)	22,021 (7.1)	1.1	22,176 (7.7)	20,406 (7.1)	−2.4
Dyslipidaemia, *n* (%)	435,784 (13.0)	353,455 (11.7)	82,329 (26.4)	38.2	78,569 (27.2)	74,344 (25.7)	−3.8
Obesity, *n* (%)	352,408 (10.5)	296,549 (9.8)	55,859 (17.9)	23.7	51,002 (17.6)	49,801 (17.2)	−1.2
Alcohol-related diagnoses, *n* (%)	184,632 (5.5)	165,522 (5.5)	19,110 (6.1)	2.8	18,056 (6.2)	17,652 (6.1)	−0.6
Chronic kidney disease, *n* (%)	116,110 (3.5)	82,155 (2.7)	33,955 (10.9)	32.9	27,889 (9.6)	28,302 (9.8)	0.6
Lung disease, *n* (%)	334,127 (10.0)	261,744 (8.6)	72,383 (23.2)	40.6	63,688 (22.0)	62,630 (21.6)	−1.0
Sleep apnoea syndrome, *n* (%)	132,626 (4.0)	107,336 (3.5)	25,290 (8.1)	19.6	22,366 (7.7)	22,248 (7.7)	−0.2
COPD, *n* (%)	182,824 (5.5)	137,856 (4.5)	44,968 (14.4)	34.1	39,484 (13.6)	38,622 (13.3)	−1.0
Liver disease, *n* (%)	112,721 (3.4)	97,796 (3.2)	14,925 (4.8)	7.9	13,259 (4.6)	13,413 (4.6)	0.3
Thyroid diseases, *n* (%)	180,643 (5.4)	138,648 (4.6)	41,995 (13.5)	31.4	32,728 (11.3)	34,529 (11.9)	2.2
Inflammatory disease, *n* (%)	176,135 (5.3)	152,159 (5.0)	23,976 (7.7)	10.9	20,727 (7.2)	20,988 (7.3)	0.4
Anaemia, *n* (%)	274,384 (8.2)	214,629 (7.1)	59,755 (19.1)	36.3	51,294 (17.7)	51,111 (17.7)	−0.2
Previous cancer, *n* (%)	500,053 (14.9)	441,098 (14.5)	58,955 (18.9)	11.7	56,123 (19.4)	55,044 (19.0)	−1.0
Poor nutrition, *n* (%)	127,089 (3.8)	97,179 (3.2)	29,910 (9.6)	26.3	25,354 (8.8)	25,439 (8.8)	0.1
Cognitive impairment, *n* (%)	112,846 (3.4)	83,072 (2.7)	29,774 (9.5)	28.6	27,068 (9.4)	26,189 (9.1)	−1.3
Frailty index, mean ± SD	6.3 ± 8.1	5.7 ± 7.7	12.2 ± 9.8	74.0	12.1 ± 10.1	11.9 ± 9.6	−2.4

* *p*-values < 0.0001 for all parameters. Abbreviations: AF—atrial fibrillation; COPD—chronic obstructive pulmonary disease; MI—myocardial infarction; PCI—percutaneous coronary intervention; SD—standard deviation; SDi—standardised difference; VF—ventricular fibrillation; VT—ventricular tachycardia.

**Table 2 jcm-12-01075-t002:** Incidence of VT/VF/CA.

Cohort	Events, (*n*)	Person-Time (Patient Years)	Incidence, (Events per 1000 Person-Years)	Incidence, %/Year (95% CI)	*p*-Value
No AF	23,252	14,587,126	1.59	0.56 (0.56–0.57)	<0.0001
AF	81,899	1,041,506	78.6	2.23 (2.20–2.26)	

Abbreviations: AF—atrial fibrillation; CA—cardiac arrest, CI—confidence interval, VF—ventricular fibrillation; VT—ventricular tachycardia.

**Table 3 jcm-12-01075-t003:** Predictors of VT/VF/CA in hospitalised patients.

Predictor	Univariable Analysis	*p* Value	Multivariable Analysis	*p* Value
	HR, 95% CI		HR, 95% CI	
Age (years)	1.038 (1.037–1.038)	<0.0001	1.023 (1.023–1.023)	<0.0001
Sex (male)	2.011 (1.986–2.036)	<0.0001	1.607 (1.586–1.628)	<0.0001
Hypertension	2.687 (2.655–2.720)	<0.0001	1.079 (1.063–1.095)	<0.0001
Diabetes mellitus	2.354 (2.321–2.386)	<0.0001	1.241 (1.222–1.261)	<0.0001
Heart failure	4.743 (4.679–4.808)	<0.0001	1.594 (1.563–1.625)	<0.0001
History of pulmonary oedema	5.007 (4.789–5.236)	<0.0001	1.269 (1.212–1.329)	<0.0001
Valve disease	3.568 (3.494–3.643)	<0.0001	1.268 (1.239–1.296)	<0.0001
Dilated cardiomyopathy	4.470 (4.369–4.574)	<0.0001	1.337 (1.303–1.373)	<0.0001
Coronary artery disease	3.392 (3.345–3.440)	<0.0001	1.221 (1.198–1.245)	<0.0001
Previous MI	3.017 (2.924–3.112)	<0.0001	1.013 (0.976–1.051)	0.50
Previous PCI	2.552 (2.489–2.617)	<0.0001	0.982 (0.953–1.011)	0.22
Vascular disease	3.230 (3.182–3.279)	<0.0001	1.232 (1.209–1.255)	<0.0001
Atrial fibrillation	3.657 (3.604–3.711)	<0.0001	1.349 (1.327–1.372)	<0.0001
Ischaemic stroke	2.064 (1.993–2.138)	<0.0001	0.966 (0.932–1.001)	0.06
Intracranial bleeding	1.821 (1.724–1.924)	<0.0001	1.006 (0.952–1.063)	0.83
Smoker	1.771 (1.737–1.805)	<0.0001	1.222 (1.196–1.248)	<0.0001
Dyslipidaemia	1.968 (1.940–1.997)	<0.0001	0.876 (0.862–0.891)	<0.0001
Obesity	1.566 (1.540–1.592)	<0.0001	1.025 (1.006–1.044)	0.009
Alcohol-related diagnoses	2.022 (1.982–2.064)	<0.0001	1.432 (1.399–1.466)	<0.0001
Chronic kidney disease	3.290 (3.222–3.360)	<0.0001	1.368 (1.337–1.399)	<0.0001
Lung disease	2.505 (2.467–2.544)	<0.0001	1.286 (1.254–1.318)	<0.0001
Sleep apnoea syndrome	1.808 (1.766–1.851)	<0.0001	1.004 (0.979–1.030)	0.74
COPD	2.912 (2.858–2.966)	<0.0001	1.015 (0.985–1.045)	0.33
Liver disease	1.996 (1.944–2.049)	<0.0001	1.171 (1.137–1.205)	<0.0001
Thyroid diseases	1.458 (1.424–1.492)	<0.0001	0.998 (0.974–1.022)	0.87
Inflammatory disease	1.177 (1.148–1.207)	<0.0001	0.910 (0.887–0.934)	<0.0001
Anaemia	2.301 (2.261–2.341)	<0.0001	1.144 (1.122–1.166)	<0.0001
Previous cancer	1.634 (1.608–1.660)	<0.0001	1.190 (1.171–1.209)	<0.0001
Poor nutrition	2.457 (2.396–2.520)	<0.0001	1.170 (1.139–1.201)	<0.0001
Cognitive impairment	2.335 (2.270–2.403)	<0.0001	0.975 (0.946–1.004)	0.09
Frailty index	1.056 (1.055–1.056)	<0.0001	1.027 (1.027–1.028)	<0.0001

Abbreviations: CA—cardiac arrest, COPD—chronic obstructive pulmonary disease; HR—hazard ratio, MI—myocardial infarction; PCI—percutaneous coronary intervention, VT—ventricular tachycardia, VF—ventricular fibrillation.

**Table 4 jcm-12-01075-t004:** Risk of VT/VF/CA in patients with and without AF.

Risk	Model A	Model B	Model C	Model D
HR, (95% CI)	HR 3.657 (3.604–3.711)	HR 2.166 (2.133–2.199)	HR 1.349 (1.327–1.372)	HR 1.339 (1.313–1.366)

Abbreviations: AF—Atrial fibrillation, CA—cardiac arrest, CI—confidence interval, HR—hazard ratio, VT—ventricular tachycardia, VF—ventricular fibrillation, cardiac arrest. Model A: Unadjusted. Model B: Adjusted for age and sex. Model C: Adjusted for all covariates in Table 1: age, sex, hypertension, diabetes mellitus, heart failure, history of pulmonary oedema, valve disease, dilated cardiomyopathy, coronary artery disease, previous MI, previous PCI, vascular disease, ischaemic stroke, intracranial bleeding, smoker, dyslipidaemia, obesity, alcohol-related diagnoses, chronic kidney disease, lung disease, sleep apnoea, COPD, liver disease, thyroid diseases, inflammatory diseases, anaemia, previous cancer, poor nutrition, cognitive impairment, frailty index. Model D: Propensity score-matched analysis.

## Data Availability

The data used in this study may be available from the corresponding author upon reasonable request.

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
