# Peer review of "Atrial Fibrillation and the Risk of Ventricular Arrhythmias and Cardiac Arrest: A Nationwide Population-Based Study"

_jcm, 2023, doi:10.3390/jcm12031075_

Round 1
Reviewer 1 Report
I congratulate the authors for conducting this very relevant study which has significant practical implications.
The association between AF and VA is known and upcoming research supports the findings from the index study. However, this large scale study does shed light into risk factors for VA amongst AF patients in a European cohort.
Specific comments:-
1) The authors must emphasise on the close association between AF and coronary artery disease (CAD) which may be an important confounding factor increasing the risk for VAs. Research has shown that coexistent CAD is quite prevalent amongst AF patients with prevalence approaching 70-82% in certain studies. I suggest the addition of these statements and citing the relevant references:-
a) Nucifora G, Schuijf JD, Tops LF, van Werkhoven JM, Kajander S, Jukema JW, Schreur JH, Heijenbrok MW, Trines SA, Gaemperli O, Turta O, Kaufmann PA, Knuuti J, Schalij MJ, Bax JJ. Prevalence of coronary artery disease assessed by multislice computed tomography coronary angiography in patients with paroxysmal or persistent atrial fibrillation. Circ Cardiovasc Imaging. 2009 Mar;2(2):100-6. doi: 10.1161/CIRCIMAGING.108.795328. Epub 2009 Jan 26. PMID: 19808575.
b) Sharma YP, Batta A, Makkar K, Hatwal J, A Gawalkar A, Kaur N, Malhi TS, Kasinadhuni G, Gupta H, Panda P, Barwad P. Angiographic profile and outcomes in persistent non-valvular atrial fibrillation: A study from tertiary care center in North India. Indian Heart J. 2022 Jan-Feb;74(1):7-12. doi: 10.1016/j.ihj.2021.12.010. Epub 2021 Dec 24. PMID: 34958796; PMCID: PMC8891025.
2) Recent research surrounding AF and VA has also shown a higher incidence of VA amongst new-onset AF patients in a large South Korean cohort. I suggest to include the same in your references:
a) Kim YG, Choi YY, Han KD, Min K, Choi HY, Shim J, Choi JI, Kim YH. Atrial fibrillation is associated with increased risk of lethal ventricular arrhythmias. Sci Rep. 2021 Sep 13;11(1):18111. doi: 10.1038/s41598-021-97335-y. PMID: 34518592; PMCID: PMC8438063.
3) An interesting finding is the increased VA in men suffering from AF compared to women. Until now we have evidence and believe that its the female gender which is more likely to develop complications in AF including stroke and dementia probably related to altered oestrogen receptor signalling in left atrial and other systemic tissues. (Include 3-4 recent references < 2 year old supporting this association). Kindly elaborate why men had increased VA in your study.
I believe this comments may help improve the overall quality of the article.
Reviewer 2 Report
I am pleased that I can review this interesting study. It covers a very large group of patients. The manuscript is generally well written, but I have some comments. How many patients had paroxysmal AF? Is there a difference in the risk for the occurrence of VT/VF/cardiac arrest between patients with paroxysmal AF and persistent or permanent AF? If the paroxysmal AF caried the same risk for VT/VF, what was the AF “burden”? Does the risk for VT/VF change after restoring sinus rhythm (drugs, eletroconversion, or catheter ablation)? I think that the authors should make some comments about these issues in their manuscript.
